# Small Airway Disease in Pulmonary Hypertension—Additional Diagnostic Value of Multiple Breath Washout and Impulse Oscillometry

**DOI:** 10.3390/jcm7120532

**Published:** 2018-12-09

**Authors:** Frederik Trinkmann, Joshua Gawlitza, Monique Künstler, Julia Schäfer, Michele Schroeter, Julia D. Michels, Ksenija Stach, Christina Dösch, Joachim Saur, Martin Borggrefe, Ibrahim Akin

**Affiliations:** 1First Department of Medicine (Cardiology, Angiology, Pulmonary and Intensive Care), University Medical Center Mannheim, 68135 Mannheim, Germany; monique.kuenstler@t-online.de (M.K.); julia.schaefer2@umm.de (J.S.); michele.schroeter@gmx.de (M.S.); julia.michels@umm.de (J.D.M.); ksenija.stach@umm.de (K.S.); christina.doesch@gmail.com (C.D.); jsaur@uni-mannheim.de (J.S.); martin.borggrefe@umm.de (M.B.); ibrahim.akin@umm.de (I.A.); 2Institute of Clinical Radiology and Nuclear Medicine, University Medical Center Mannheim, 68135 Mannheim, Germany; joshua.gawlitza@umm.de; 3DZHK (German Center for Cardiovascular Research), Partner Site Mannheim, University Medical Center Mannheim, 68135 Mannheim, Germany

**Keywords:** multiple breath washout, impulse oscillometry, lung clearance index, small airway disease, pulmonary hypertension

## Abstract

Airways obstruction is frequent in patients with pulmonary hypertension (PH). Small airway disease (SAD) was identified as a major contributor to resistance and symptoms. However, it is easily missed using current diagnostic approaches. We aimed to evaluate more elaborate diagnostic tests such as impulse oscillometry (IOS) and SF_6_-multiple-breath-washout (MBW) for the assessment of SAD in PH. Twenty-five PH patients undergoing body-plethysmography, IOS and MBW testing were prospectively included and equally matched to pulmonary healthy and non-healthy controls. Lung clearance index (LCI) and acinar ventilation heterogeneity (S_acin_) differed significantly between PH, healthy and non-healthy controls. Likewise, differences were found for all IOS parameters between PH and healthy, but not non-healthy controls. Transfer factor corrected for ventilated alveolar volume (TLCO/VA), frequency dependency of resistance (D5-20), resonance frequency (F_res_) and S_acin_ allowed complete differentiation between PH and healthy controls (AUC (area under the curve) = 1.0). Likewise, PH patients were separated from non-healthy controls (AUC 0.762) by D5-20, LCI and conductive ventilation heterogeneity (S_cond_). Maximal expiratory flow (MEF) values were not associated with additional diagnostic values. MBW and IOS are feasible in PH patients both providing additional information. This can be used to discriminate PH from healthy and non-healthy controls. Therefore, further research targeting SAD in PH and evaluation of therapeutic implications is justified.

## 1. Introduction

Pulmonary hypertension (PH) is most frequently associated with left heart (Nizza Group 2) or pulmonary disease (Group 3). Management of PH in these groups is typically limited to treatment of the underlying disease [1]. In contrast, targeted therapy is available for Group 1 and 4 PH that is directly attributable to vascular changes and has dramatically improved in recent years. Irrespective of the underlying mechanism, dyspnea is almost universally reported by PH patients being a complex symptom potentially caused by cardiovascular, respiratory and neuromuscular abnormalities [2,3]. While changes in the cardiovascular system are well investigated [4,5], little is known about the underlying mechanisms for affection of adjacent airways and pulmonary structures. Airway obstruction as assessed by spirometry is often present in patients with pulmonary arterial hypertension (PAH) [6,7]. Although restrictive ventilation disorders are frequently described in Group 2 PH [8], obstruction patterns are also common in patients with left heart failure [9]. Although these airway abnormalities may often be viewed as mild at rest, they were shown to induce dynamic hyperinflation [10]. They may contribute to symptoms during physical stress and eventually reduce exercise capacity. This does not only hold for Group 1 PH. A morphological overlap between lesions found in Groups 1 and 3 was previously described [11,12] and may imply similar changes in the small airways, irrespective of the underlying cause of PH. In PAH, peripheral airways were identified as the major site of obstruction [7,13]. However, small airway disease can be easily missed with commonly used diagnostic tests, most notably spirometry. More advanced lung function testing includes body-plethysmography and determination of transfer factor, but also novel techniques such as impulse oscillometry (IOS) or multiple breath washout (MBW) testing. IOS has been shown to detect small airway disease in chronic obstructive pulmonary disease (COPD) and relates to symptoms especially dyspnea [14]. Likewise, affection of small airways can also be found in stable ischemic heart disease [15]. MBW yielded promising results for the determination of ventilation heterogeneity in patients with COPD. Lung clearance index (LCI) is already elevated even in the absence of spirometric obstruction [16]. Both techniques may therefore provide additional information when assessing lung function impairment in patients with PH. However, it is still unknown whether bronchodilator therapy can improve symptoms in this setting [13,17]. This is not only important in Groups 2, 3 and 5 PH where targeted therapy is unavailable. In Group 1 PH, escalation of targeted therapy often has to be balanced against costs and undesirable side effects that could potentially be avoided with anti-obstructive treatment. With many patients remaining symptomatic despite maximal treatment, affection of small airways missed by conventional lung function testing may provide a missing link. The aim of the present study, therefore, was to prospectively evaluate novel lung function tests for assessment of small airway disease in patients with PH. 

## 2. Experimental Section

### 2.1. Subjects

We prospectively evaluated patients with known or first diagnosis of PH who were matched to pulmonary healthy and non-healthy controls. Written informed consent was obtained from all participants prior to inclusion. The study protocol was approved by our local ethics committee, compliant with the Declaration of Helsinki and registered at clinicaltrials.gov (NCT03667794). Healthy controls had normal lung function testing including body-plethysmography and transfer factor, no previously diagnosed pulmonary disease, as well as no respiratory symptoms. Non-healthy controls were allowed to have stable pulmonary comorbidities including COPD, sarcoidosis, asthma, or fibrosis as well as non-pulmonary comorbidities. Lung function testing including the shapes of flow-volume and flow-pressure curves was independently assessed by two experienced investigators. Patients in unstable clinical condition or suffering from infective lung disease were not included.

### 2.2. Study Protocol

All subjects underwent three consecutive MBW tests in upright position followed by IOS (MasterScreen IOS, CareFusion 234 GmbH, Höchberg, Germany) and whole-body plethysmography (MasterScreen Body). Functional residual capacity (FRC) was determined from end-expiratory shutter maneuvers during normal breathing. Transfer factor corrected for ventilated alveolar volume (TLCO/VA) was determined in single breath technique. If obstruction was present (FEV_1_ (forced expiratory volume in one second)/VC (vital capacity) <80% of predicted), we performed reversibility testing with doses of 40 µg ipratropium bromide and 100 µg fenoterol hydrobromide administered via soft-mist haler (Berodual Respimat, Boehringer Ingelheim Pharma GmbH & Co. KG, Germany). Reversibility was assessed separately according to an increasing FEV_1_, decreasing residual volume (RV) or decreasing area under reactance curve (AX) for spirometry, body plethysmography and IOS, respectively. Diagnostic work-up in patients with PH was performed according to current ESC/ERS (European Society of Cardiology/European Respiratory Society) guidelines [1]. Transthoracic echocardiography was used for screening followed by invasive confirmation during right heart catheterization. Hemodynamic data was then collected from the digital patient record. A commercially available closed-circuit system (Innocor, PulmoTrace ApS, Glamsbjerg, Denmark) was used for MBW measurements as previously described in detail [18]. The device consists of a 3-liter rebreathing bag filled with a mixture of room air and test gas (94% O_2_, 1% SF_6_ and 5% N_2_O, PulmoTrace ApS) from an on-board gas cylinder. FRC, LCI, acinar (S_acin_) and conductive (S_cond_) ventilation heterogeneity were derived from three consecutive wash-outs using proprietary software provided by the manufacturer (software version 8.0 beta 1). Subjects were breathing tidally, and the test was stopped when end tidal SF_6_ had fallen below 1/40 of the starting concentration. Only patients with at least two technically acceptable LCI measurements based on slightly modified ATS/ERS (American Thoracic Society/European Respiratory Society) criteria (online Appendix A) were included in the final analysis. 

### 2.3. Statistical Analysis

Mean values are given ± standard deviation (SD) unless stated otherwise. Differences between groups were assessed by Student’s *t*-test for continuous variables or Chi-squared test for categorical variables. ANOVA (Analysis of variance) adjusted for multiple testing by Tukey HSD was used to reveal differences between types of PH groups. The coefficient of variation (CV) was calculated as SD/mean from the valid MBW measurements. We calculated that a planned sample size of 23 per group would provide 80% power for detecting a difference of 1.0 ± 1.6 in LCI and 20 ± 32 percentage points in D5-20, respectively. An alpha error of less than 5% in two-sided testing was considered statistically significant. R Statistical Software (v3.4.2, Foundation for Statistical Computing, Vienna, Austria) was used for all data analysis [19]. Propensity score matching was performed with a 1:1 ratio and nearest neighbor approach (online supplement) to find adequate healthy as well as non-healthy controls, respectively [20]. Inherent differences were introduced intentionally between healthy and non-healthy controls. Therefore, unpaired tests were used for an individual comparison of PH patient to the respective group. Diagnostic performance was evaluated using receiver operating curve (ROC) analysis and calculation of area under the curve (AUC). All lung function parameters with a highly significant difference (*p* < 0.01) between PH patients and healthy controls in univariate analysis were included in a stepwise generalized linear model. For assessment of discriminative capability between PH patients and non-healthy controls, we included all parameters of IOS and MBW. A stepwise multiple linear regression model was used to evaluate the influence of comorbidities (COPD, asthma, sarcoidosis, fibrosis, pulmonary hypertension) and anthropometric parameters (gender, age, height, weight, smoking status) on LCI and D5-20, respectively. 

## 3. Results

Final analysis was performed in 75 datasets with baseline characteristics given in Table 1. After matching patients with PH and non-healthy controls, no differences in comorbidities were found between groups for COPD (7 vs. 8), bronchial asthma (2 vs. 2), sarcoidosis (5 vs. 6) and fibrosis (2 vs. 3). Reversibility testing was performed in 12 patients (48%) with PH and 9 non-healthy controls (36%) with details given in Appendix A.

At least two successful LCI measurements were obtained in all subjects. Slope analysis was not possible in four (16%) patients of the PH and four healthy controls each (*p* = 0.72) as well as in eight (32%) non-healthy controls (*p* = 0.38, Chi-squared test, Appendix A). Mean LCI differed significantly between patients with PH (8.7 ± 1.3) and their respective healthy (7.4 ± 0.8, *p* < 0.001) as well as non-healthy (10.1 ± 2.9, *p* < 0.05) controls. Likewise, differences were found for all IOS parameters between patients with PH and healthy but not non-healthy controls (Table 2). S_acin_ differed significantly only between patients with PH and healthy controls (0.26 ± 0.12 vs. 0.1 ± 0.06, *p* < 0.00001) whereas differences in S_cond_ were only found as compared to non-healthy controls (0.03 ± 0.07 vs. 0.06 ± 0.03, *p* < 0.05). CV for LCI was 3.5 ± 2.2% in PH patients, 2.8 ± 2.1% in healthy controls (*p* = 0.25) and 3.5 ± 2.5% in non-healthy controls (*p* > 0.9). PH and age remained as independent predictors for LCI in stepwise multiple linear regression (adjusted *R*^2^ = 0.30, *p* < 0.0001) being associated with a 0.96 increase if present and an 0.18 increase per decade, respectively. Presence of COPD, sarcoidosis, fibrosis and PH as well as height and weight remained as predictors for D5-20 (adjusted *R*^2^ = 0.47, *p* < 0.00001).

Hemodynamic data and Nizza classification of patients with PH (*n* = 25) are summarized in Table 3. The most frequent comorbidities in patients with PH were arterial hypertension (68%), atrial fibrillation (56%) and coronary heart disease (44%) as given in Appendix A. No meaningful correlations were found between mean pulmonary arterial pressure (mPAP) and any of the MBW (highest correlations: S_acin_
*r* = −0.23, *p* = 0.33), IOS (F_res_
*r* = 0.05, *p* = 0.82) as well as conventional lung function parameters (VC *r* = −0.18, *p* = 0.43). No differences were found in LCI between patients with pre-capillary (8.3 ± 1.2), post-capillary (8.8 ± 1.3) and mixed PH (9.3 ± 1.5) in ANOVA (*p* = 0.92). In contrast, patients with mixed PH had higher S_acin_ of 0.35 ± 0.11 as compared to post-capillary (0.26 ± 0.13, adjusted *p* = 0.36) and pre-capillary (0.19 ± 0.05, adjusted *p* = 0.07) whereas no differences were found for S_cond_ (pre 0.05 ± 0.08, post 0.04 ± 0.05, mixed 0.0 ± 0.09, *p* = 0.94, ANOVA). 

Including TLCO/VA, D5-20, F_res_ and S_acin_ in a generalized linear model allowed complete differentiation between PH patients and healthy controls (AUC = 1.0, Figure 1A,B). Likewise, PH patients could be separated from non-healthy controls when including D5-20, LCI and S_cond_ in the model with an AUC of 0.762 whereas S_acin_, F_res_ and AX did not provide additional information (Figure 1C,D). Maximal expiratory flow (MEF) at 75%, 50% and 25% of vital capacity (VC) was also not associated with additional diagnostic value (all AUC <0.6, Appendix A).

## 4. Discussion

We were able to demonstrate that affection of small airways can be evaluated using novel lung function tests in patients with PH. LCI is significantly increased in PH compared to healthy controls with normal lung function and significantly lower compared to non-healthy controls with comparable impairments in conventional lung function testing. Non-healthy controls and PH patients differed in conductive but not acinar ventilation heterogeneity. In contrast, acinar ventilation heterogeneity was increased in PH compared to healthy controls. For all IOS parameters, significant differences were seen between PH patients and healthy but not non-healthy controls. In accordance with previous results, we found signs of hyperinflation indicated by an increased RV/TLC as well as obstructive lung disease indicated by a reduced FEV_1_/VC. However, differences in the investigated collectives should be noted. While Meyer and co-workers focused on primary pulmonary hypertension (similarly today’s Nizza Groups 1 and 4) [13], we also included patients with Nizza Groups 2 and 3. This led to a substantially older mean age especially in the PH group as well as a more balanced gender distribution. We decided to also include these groups in our analysis in order to give a realistic representation of patients with PH to be investigated with IOS and MBW in clinical routine as well as in scientific context. Additionally, Group 2 PH is the clinically most relevant group and contains the most frequent differential diagnosis. As stated above, morphological similarity between lesions found in Groups 1 and 3 may cause similar changes in the small airways, irrespective of the underlying cause of PH. Moreover, patients with known lung disease were included in our cohort, all known to also influence parameters of small airway disease. When correcting for these factors in our multivariate model, only PH and age remained as independent predictors for LCI as compared to healthy controls. Previous research suggests that age is the only relevant anthropometric factor influencing LCI in adults. We found a 0.18 increase in LCI per decade which compares well to the 0.22 found by Verbanck and co-workers [21]. Until today, solid reference values for adult SF_6_-MBW are lacking. However, age seems to be the most relevant contributor to nitrogen (N_2_) based MBW measurement of global as well as local ventilation heterogeneity in adulthood. The influence of height seems to be negligible in patients older than 6 years while there is a nonlinear decrease pattern when using SF_6_ as tracer gas [22]. For D5-20, presence of obstructive as well as restrictive lung disease, height and weight remained as predictors in multivariate testing. PH patients were additionally compared to a non-healthy control group matched for pulmonary disease, impairment of lung function as well as anthropometric parameters. While no differences were found for IOS parameters, LCI and S_cond_ further increased as compared to the PH group. Combining MBW and IOS parameters contains additional information to conventional lung function testing. A combination of LCI, S_cond_ and D5-20 allowed a reasonable discrimination between patients with PH and non-healthy controls although both groups were comparable in conventional lung function testing. Likewise, novel parameters together with TLCO/VA allowed complete discrimination between PH patients and healthy controls that outperformed all conventional lung function parameters. Although MEF values were previously shown to be altered in primary pulmonary hypertension [13], they did not provide additional diagnostic information in our investigation. This is an important finding as MEF and derived parameters are still used as surrogates for small airway disease in clinical routine as well as medical education. Although not reaching statistical significance, reversibility criteria were more likely to be met for AX and RV than for FEV_1_ in the PH group. No differences were found for LCI and S_cond_ between the type of PH. However, patients with mixed PH showed higher values for S_acin_ as compared to post-capillary and pre-capillary PH that may be of clinical value despite marginally missing statistical significance. This should be further considered when evaluating the diagnostic performance of novel and conventional lung function testing. In COPD patients, it was demonstrated that small airway disease considerably contributes to dyspnea [14]. Likewise, small airway disease without airflow limitation in conventional lung function testing was frequently found in smokers with ischemic heart disease and associated with poorer clinical condition and greater future cardiac risk [15]. Heart failure due to ischemic heart disease is associated with post-capillary pulmonary hypertension and may potentiate detrimental effects of small airway disease. Therefore, it may also be a potential therapeutic target for patients suffering from PH justifying future interventional trials. Management of comorbidities is crucial but can also be challenging. Previously, substantial structural improvements were demonstrated by introduction of care manager nurses. Becoming more adherent to testing and treatment recommendations had the potential to reduce hospitalizations and emergency care costs. Moreover, satisfaction among patients, physicians, and care managers was increased while the concept cannot only be transferred to practice settings but also include home visits [23]. Care managers could therefore also improve management of patients with PH irrespectively of the underlying cause.

When assessing our results, several technical aspects should be taken into consideration. Forced maneuvers such as FEV_1_ require patient collaboration whereas IOS and MBW are based on tidal breathing. Both techniques were demonstrated to be feasible in PH patients in the study at hand. At least two successful LCI measurements could be obtained in all subgroups in a reasonable time frame which corresponds well to previous findings in patients with COPD, bronchial asthma and sarcoidosis using SF_6_-MBW [24]. In contrast, durations of up to 20 minutes have been reported for a single measurement in patients with severe COPD using a N_2_-based setup leading to a considerably lower rate success rate of 55% in these patients [25]. When assessing local ventilation heterogeneity, S_acin_ and S_cond_ could not be derived in 16% of PH patients and healthy controls as well as 32% of non-healthy controls. This corresponds to considerably larger CVs shown for these parameters [26] and postulations of lower success rates as a result of the more elaborate underlying algorithm. Moreover, meaningful differences in MBW outcome measures have been described [27,28]. Due to the direct measurement, SF_6_-based setups are being considered the gold standard from metrological considerations [29]. In N_2_--based approaches, inaccuracies are introduced by the indirect measurement technique and N_2_ back diffusion is a major issue that cannot be reliably corrected for yet [30,31]. Consequently, attention has to be paid to inert gas choice when interpreting MBW results.

Overall statistical power was sufficient to detect between-group differences. The a priori estimated differences were met except for D5-20 when comparing non-healthy controls to PH patients. Nevertheless, it should be noted that this does not necessarily hold for subgroups such as subtypes of PH. Therefore, further research is warranted targeting specific subgroups, most notably specifically treatable Nizza Groups 1 and 4. Secondly, right heart catherization and echocardiography data were taken in a rather retrospective manner from patient records. Although this may affect interpretation of lung function testing in context of these parameters, we do not consider it a major drawback of our study as it was primarily powered to detect differences in lung function parameters and their predictive potential. In accordance with previous findings, we were not able to detect correlations between conventional lung function testing and hemodynamics [7]. With the abovementioned restrictions, we found neither correlations of the novel parameters. Thirdly, pharmacological history was not available consistently throughout the whole collective. Therefore, we refrained from analysis of these data as non-randomly distributed missing data may have introduced a systematic bias. Finally, our rigorous definition of pulmonary healthy controls made identification challenging over the whole age range. As a result, statistically significant differences were found for age and height for this group as compared to older PH patients. However, the two-decade difference in age does not explain e.g., the difference in LCI which clearly exceeds the expected 0.4 change. An overproportionate increase was found in the equally old non-healthy controls for LCI that is also not attributable to age-effects. Therefore, we do not consider this to limit discriminative or predictive power of our study in general.

## 5. Conclusions

Novel lung function tests were shown to be feasible in patients with PH. Statistically significant differences were found for both global and local parameters of ventilation heterogeneity derived from MBW as well as from resistance measurements using IOS. Both techniques provide additional information not only in discriminating patients with PH from healthy but also non-healthy controls. Based on our findings, further research is justified targeting small airway disease in patients with PH and evaluation of therapeutic implications.

## Figures and Tables

**Figure 1 jcm-07-00532-f001:**
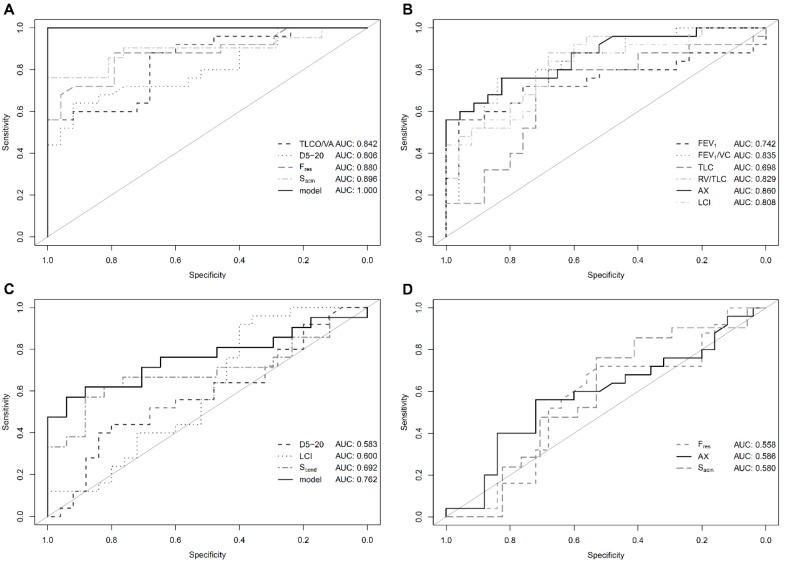
Diagnostic performance: Receiver operating curve (ROC) analysis for conventional and novel lung function parameters in patients with pulmonary hypertension (PH) vs. healthy controls (**A**,**B**) as well as patients with PH vs. non-healthy controls (**C**,**D**). Solid black lines indicate best performing parameter of the respective panel. Abbreviations are identical with Table 2. (**A**) Overall good performance of individual parameters selected for inclusion in the generalized linear model. (**B**) Parameters not selected for inclusion in the generalized linear model. (**C**) Generalized linear model with improved diagnostic performance as compared to individual parameters. (**D**) Parameters not selected for inclusion in the generalized linear model.

**Table 1 jcm-07-00532-t001:** Baseline characteristics.

		PH (*n* = 25)		Healthy Controls (*n* = 25)			Non-Healthy Controls (*n* = 25)			
	Unit	Mean	Range	Mean	Range	*p*-Value ^#^	Mean	Range		*p*-Value ^#^
Age	years	73 ± 10	46–86	55 ± 18	22–85	<0.001 *	72 ± 13	40–86	0.65	
Male		13 (52%)		15 (60%)		0.78	14 (56%)		>0.9	
Weight	kg	76 ± 16	53–110	81 ± 22	51–132	0.37	76 ± 16	53–110	0.98	
Height	cm	164 ± 9	149–178	171 ± 9	157–198	0.01 *	165 ± 7	154–180	0.51	
Obesity										
yes/no	*n*	8/17		8/17		>0.9	9/16		>0.9	
	%	32/68		32/68			36/64			
Arterial hypertension										
yes/no	*n*	17/8		4/21		<0.001 *	16/9		>0.9	
	%	68/32		16/84			64/36			
Diabetes mellitus										
yes/no	*n*	7/18		2/23		0.14	4/21		0.49	
	%	28/72		8/92			16/84			
Smoker										
yes/ex/no	*n*	2/10/13		3/6/16		0.47	3/11/11		0.81	
	%	8/40/52		12/24/64			12/44/44			

kg: Kilogram, cm: Centimeter. ^#^ Student’s *t*-test as compared to the PH group. Obesity was defined as body mass index (BMI) ≥30 kg/m^2^. * Statistically significant *p* < 0.05.

**Table 2 jcm-07-00532-t002:** Lung function data.

		PH (*n* = 25)		Healthy Controls (*n* = 25)					Non-Healthy Controls (*n* = 25)		
	Unit	Mean	Range	Mean		Range		*p*-Value ^#^	Mean	Range	*p*-Value ^#^
**Spirometry**											
FEV_1_/VC	%	88 ± 13	44–109		98 ± 8		84–115	<0.01 *	86 ± 19	38–116	0.61
FEV_1_	%pred	81 ± 32	27–152		101 ± 16		65–135	0.01 *	74 ± 27	30–135	0.41
VC	%pred	91 ± 26	44–147		102 ± 16		64–124	0.06	88 ± 27	37–135	0.75
**Body Plethysmography**											
TLC	%pred	97 ± 15	63–125		107 ± 10		87–124	0.01 *	102 ± 31	51–166	0.47
RV	%pred	119 ± 25	80–188		120 ± 12		82–142	0.78	130 ± 57	56–324	0.38
RV/TLC	%	51 ± 9	33–71		39 ± 8		27–57	<0.00001 *	52 ± 11	30–83	0.88
FRC_pleth_	L	3.1 ± 0.7	1.8–4.5		3.1 ± 0.5		2.4–4.2	0.64	3.3 ± 1.2	1.8–6.2	0.30
TLCO/VA	%pred	69± 22	18–103		96 ± 10		80–115	<0.00001 *	70 ± 29	12–115	0.89
**Impulse Oscillometry**											
D5-20	%	49 ± 36	5–114		14 ± 12		0–47	<0.0001 *	40 ± 39	0–170	0.39
F_res_	Hz	19 ± 6	9–37		11 ± 4		3–19	<0.00001 *	19 ± 7	8–35	0.74
AX	-	1.58 ± 1.62	0.11–7.8		0.29 ± 0.27		0.01–1.1	<0.001 *	1.30 ± 1.66	0.0–7.0	0.55
**Multiple Breath Washout**											
LCI	-	8.7 ± 1.3	6.7–11.6		7.4 ± 0.8		6.2–8.9	<0.001 *	10.1 ± 2.9	7.2–17.6	0.04 *
FRC_MBW_	L	2.6 ± 0.8	1.2–4.4		2.9 ± 0.8		1.2–4.4	0.14	2.5 ± 0.6	1.3–4.3	0.53
S_acin_	L^−1^	0.26 ± 0.12	0.03–0.46		0.1 ± 0.06		−0.04–0.18	<0.00001 *	0.32 ± 0.35	−0.14–1.12	0.50
S_cond_	L^−1^	0.03 ± 0.07	−0.14–0.15		0.05 ± 0.04		−0.05–0.13	0.24	0.06 ± 0.03	0.0–0.13	0.04 *

FEV_1_: Forced expiratory volume in one second, VC: Vital capacity, TLC: Total lung capacity, RV: Residual volume, FRC_pleth/_: Functional residual capacity by body plethysmography, FRC_MBW_: Functional residual capacity by multiple breath washout, TLCO/VA: Transfer factor corrected for ventilated alveolar volume, D5-20: Frequency dependence of resistance, F_res_: Resonance frequency, AX: Area under reactance curve, LCI: Lung clearance index, S_acin_: Acinar ventilation heterogeneity, S_acin_: Conductive ventilation heterogeneity, %pred: percent of predicted. ^#^ Student’s *t*-test as compared to the PH group. * Statistically significant *p* < 0.05.

**Table 3 jcm-07-00532-t003:** Hemodynamic data.

		PH (*n* = 25)	
	Unit	Mean	Range
**Right Heart Catheter**			
mPAP	mmHg	34 ± 14	22–87
PAWP	mmHg	16 ± 6	2–25
DPG	mmHg	4 ± 8	−6–24
type	*n*	10/11/4	
pre/post/mixed	%	40/44/16	
Nizza class	*n*	7/13/2/2/1	
1/2/3/4/5	%	28/52/8/8/4	
**Echocardiography**			
sPAP	mmHg	57 ± 19	20–90
TAPSE	mm	19 ± 4	13–28
heart failure	*n*	2/13/4	
sys/dia/both	%	8/52/16	

mPAP: Mean pulmonary arterial pressure, PAWP: Pulmonary arterial wedge pressure, DPG: Diastolic pressure gradient, pre: Pre-capillary, post: Post-capillary, sPAP: Systolic pulmonary arterial pressure, TAPSE: Tricuspid annular plane systolic excursion, sys: Systolic, dia: Diastolic.

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
