# Peer review of "Small Airway Disease in Pulmonary Hypertension—Additional Diagnostic Value of Multiple Breath Washout and Impulse Oscillometry"

_jcm, 2018, doi:10.3390/jcm7120532_

Reviewer 1 Report

This is a manuscript describing diagnostic efficacy of IOS and MBW in patients with PH.

I have two comments.

1)   Patients with PH included COPD, sarcoidosis, bronchial asthma and fibrosis (Table e3). The data should be analyzed only in group 1 PH.  

2)   Data were analyzed using student's t-test in Table 1 and 2.  However, these data should be analyzed using an appropriate analysis of a variance model followed by an appropriate post hoc test.

Author Response

Point-by-point answers

We would like to thank both reviewers for the expeditious evaluation and helpful comments on our manuscript. We thoroughly revised the article according to their recommendations and think this substantially enhanced its quality. Specifically, description of the methods and clarity of results were improved as suggested. Please find enclosed our point-to-point answers as well as the revised document. Of course, we stand ready to make additional improvements as the reviewers require.

Reviewer 1

Patients with PH included COPD, sarcoidosis, bronchial asthma and fibrosis (Table e3). The data should be analyzed only in group 1 PH.  

We agree that group 1 PH is of outstanding interest and pathophysiologically well described. However, we decided to also include the clinically relevant groups 2 and 3 in our analysis. This gives a realistic representation of patients with PH to be investigated with IOS and MBW in clinical routine as well as in scientific context. Moreover, group 2 PH is the clinically most relevant group and contains the most frequent differential diagnosis. It should further be noted that a morphological overlap between lesions found in group 1 and group 3 PH was previously described (Seeger W et al. Pulmonary hypertension in chronic lung diseases. J Am Coll Cardiol. 2013). This may imply similar changes in the small airways in both groups that can be detected using IOS and MBW, irrespective of the underlying cause of PH. Concomitant disease such as COPD, sarcoidosis, bronchial asthma and fibrosis may nevertheless lead to confounding (please also see our answer to reviewer 2 below). We controlled for this during matching resulting in a balanced distribution of comorbidities in the PH groups and non-healthy controls.

Data were analyzed using student's t-test in Table 1 and 2.  However, these data should be analyzed using an appropriate analysis of a variance model followed by an appropriate post hoc test.

Of course, we agree that correcting for multiple testing is an important issue in multivariate analysis. We decided for a group-based comparison by design. PH patients were compared to either healthy or non-healthy controls in pairs. Due to our matching process (please also see our answer to reviewer 2 below) the two control groups can be considered different and hence no test correction is required. Therefore, PH patients were compared to the respective control group using a t-distribution rather than f-distribution based test separately. In contrast, intra-group differences were compared using ANOVA and p-values were corrected for multiple testing using Tukey HSD. A respective statement was added to the “Statistical analysis” section and of course we stand ready for further queries as required.

Reviewer 2

The authors should specify how the patients were diagnosed as suffering from PH. Please provide.

Diagnostic work-up was performed according to current ESC/ERS guidelines and a respective statement was added to the “Study protocol” section.

All cardiovascular risk factors should be included in the table. Please update table 1.

We thankfully acknowledge this important point and added information on cardiovascular risk factors to Table 1 as proposed.

Pharmacological history of the patients should be included and discussed in the final analysis.

We have to apologize that pharmacological history was not able consistently throughout the whole collective. While medication of PH patients was well documented in general this does not hold for the others groups. This would generate non-randomly distributed missing data and finally introduce a systematic bias. Therefore, we refrained from presentation and analysis of these data. A respective paragraph was added to the “Limitations” section of the discussion.

A multivariate regression analysis should be performed in order to evaluate the role of confounding factors on final results.

We agree that confounding is important to be controlled for. We primarily chose an a priori approach for this and both control groups were matched for commonly known factors. Specifically, anthropometric parameters such as height, weight and gender affect lung function parameters. Solid reference values correcting for this are available for spirometry, body plethysmography. Therefore, confounding of these factors can be reliably excluded. In contrast, normative data is sparse for MBW especially when using SF6 as tracer gas. When correcting for factors in our multivariate model, only presence PH and age remained as independent predictors for LCI as compared to healthy controls. This is in accordance with previous research that suggests that age is the only relevant anthropometric factor influencing LCI in adults as outlined in the “Discussion” section. Due to our sample size, further addition of factors would potentially violate test assumptions of the underlying model. As a consequence, generalizability would have been negatively affected. However, we want to point out that significant differences for baseline characteristics were only found for arterial hypertension between PH patients and healthy controls. For all other potentially influencing factors either no difference was found or we were able to control for confounding in advance.

For D5-20, presence of obstructive as well as restrictive lung disease, height and weight remained as predictors in multivariate testing. The latter two do not seem to be problematic as D5-20 is given as percentage value which is comparable between individuals. As outlined in the “Limitations” section, our rigorous definition of pulmonary healthy controls made identification challenging over the whole age range. This resulted in statistically significant differences for age and height for this group as compared to PH patients. However, this two-decade difference in age does not explain e.g. the difference in LCI which clearly exceeds the 0.4 change that can be estimated from our as well as previous findings. An overproportionate increase was found in the equally old non-healthy controls for LCI that is also not attributable to age-effects. In summary, we do not consider this to limit discriminative or predictive power of our study in general and of course we stand ready for further queries if required.

The small sample size can influence results. Please discuss such a point in a dedicated limitation section.

We agree that sample size can have considerable impact on the results found in our study. Therefore, a priori sample size calculation was performed as outlined in the “Statistical analysis” section. Based on this, actual sample sizes of 25 in each group provide at least 80% power to detect a 1.0 difference in LCI while actual mean differences were even higher with 1.3 and 1.4 as compared to the healthy and non-healthy control groups, respectively. An estimated 20 percentage point difference was not reached exclusively for D5-20 when comparing non-healthy controls to PH patients. In contrast, there was an actual 35 percentage point difference between healthy controls and PH patients. The “Limitations” section in the discussion was revised accordingly. 

The role of care manager should be discussed. Please consider and discuss the paper from Ciccone MM et al. Vasc Health Risk Manag. 2010 May 6;6:297-305.

A respective paragraph was added to the “Discussion” section as proposed and we want to thank you for introducing this important aspect.

Reviewer 2 Report

To:

Editorial Board

Journal of Clinical Medicine

Title: “Small airway disease in pulmonary hypertension – additional diagnostic value of multiple breath washout and impulse oscillometry”.

Dear Editor,

I read this manuscript and I think that:

-          The authors should specify how the patients were diagnosed as suffering from PH. Please provide.

-          All cardiovascular risk factors should be included in the table. Please update table 1.

-          Pharmacological history of the patients should be included and discussed in the final analysis.

-          A multivariate regression analysis should be performed in order to evaluate the role of confounding factors on final results.

-          The small sample size can influence results. Please discuss such a point in a dedicated limitation section.

-          The role of care manager should be discussed. Please consider and discuss the paper from Ciccone MM et al. Vasc Health Risk Manag. 2010 May 6;6:297-305.

Author Response

Point-by-point answers

We would like to thank both reviewers for the expeditious evaluation and helpful comments on our manuscript. We thoroughly revised the article according to their recommendations and think this substantially enhanced its quality. Specifically, description of the methods and clarity of results were improved as suggested. Please find enclosed our point-to-point answers as well as the revised document. Of course, we stand ready to make additional improvements as the reviewers require.

Reviewer 1

Patients with PH included COPD, sarcoidosis, bronchial asthma and fibrosis (Table e3). The data should be analyzed only in group 1 PH.  

We agree that group 1 PH is of outstanding interest and pathophysiologically well described. However, we decided to also include the clinically relevant groups 2 and 3 in our analysis. This gives a realistic representation of patients with PH to be investigated with IOS and MBW in clinical routine as well as in scientific context. Moreover, group 2 PH is the clinically most relevant group and contains the most frequent differential diagnosis. It should further be noted that a morphological overlap between lesions found in group 1 and group 3 PH was previously described (Seeger W et al. Pulmonary hypertension in chronic lung diseases. J Am Coll Cardiol. 2013). This may imply similar changes in the small airways in both groups that can be detected using IOS and MBW, irrespective of the underlying cause of PH. Concomitant disease such as COPD, sarcoidosis, bronchial asthma and fibrosis may nevertheless lead to confounding (please also see our answer to reviewer 2 below). We controlled for this during matching resulting in a balanced distribution of comorbidities in the PH groups and non-healthy controls.

Data were analyzed using student's t-test in Table 1 and 2.  However, these data should be analyzed using an appropriate analysis of a variance model followed by an appropriate post hoc test.

Of course, we agree that correcting for multiple testing is an important issue in multivariate analysis. We decided for a group-based comparison by design. PH patients were compared to either healthy or non-healthy controls in pairs. Due to our matching process (please also see our answer to reviewer 2 below) the two control groups can be considered different and hence no test correction is required. Therefore, PH patients were compared to the respective control group using a t-distribution rather than f-distribution based test separately. In contrast, intra-group differences were compared using ANOVA and p-values were corrected for multiple testing using Tukey HSD. A respective statement was added to the “Statistical analysis” section and of course we stand ready for further queries as required.

Reviewer 2

The authors should specify how the patients were diagnosed as suffering from PH. Please provide.

Diagnostic work-up was performed according to current ESC/ERS guidelines and a respective statement was added to the “Study protocol” section.

All cardiovascular risk factors should be included in the table. Please update table 1.

We thankfully acknowledge this important point and added information on cardiovascular risk factors to Table 1 as proposed.

Pharmacological history of the patients should be included and discussed in the final analysis.

We have to apologize that pharmacological history was not able consistently throughout the whole collective. While medication of PH patients was well documented in general this does not hold for the others groups. This would generate non-randomly distributed missing data and finally introduce a systematic bias. Therefore, we refrained from presentation and analysis of these data. A respective paragraph was added to the “Limitations” section of the discussion.

A multivariate regression analysis should be performed in order to evaluate the role of confounding factors on final results.

We agree that confounding is important to be controlled for. We primarily chose an a priori approach for this and both control groups were matched for commonly known factors. Specifically, anthropometric parameters such as height, weight and gender affect lung function parameters. Solid reference values correcting for this are available for spirometry, body plethysmography. Therefore, confounding of these factors can be reliably excluded. In contrast, normative data is sparse for MBW especially when using SF6 as tracer gas. When correcting for factors in our multivariate model, only presence PH and age remained as independent predictors for LCI as compared to healthy controls. This is in accordance with previous research that suggests that age is the only relevant anthropometric factor influencing LCI in adults as outlined in the “Discussion” section. Due to our sample size, further addition of factors would potentially violate test assumptions of the underlying model. As a consequence, generalizability would have been negatively affected. However, we want to point out that significant differences for baseline characteristics were only found for arterial hypertension between PH patients and healthy controls. For all other potentially influencing factors either no difference was found or we were able to control for confounding in advance.

For D5-20, presence of obstructive as well as restrictive lung disease, height and weight remained as predictors in multivariate testing. The latter two do not seem to be problematic as D5-20 is given as percentage value which is comparable between individuals. As outlined in the “Limitations” section, our rigorous definition of pulmonary healthy controls made identification challenging over the whole age range. This resulted in statistically significant differences for age and height for this group as compared to PH patients. However, this two-decade difference in age does not explain e.g. the difference in LCI which clearly exceeds the 0.4 change that can be estimated from our as well as previous findings. An overproportionate increase was found in the equally old non-healthy controls for LCI that is also not attributable to age-effects. In summary, we do not consider this to limit discriminative or predictive power of our study in general and of course we stand ready for further queries if required.

The small sample size can influence results. Please discuss such a point in a dedicated limitation section.

We agree that sample size can have considerable impact on the results found in our study. Therefore, a priori sample size calculation was performed as outlined in the “Statistical analysis” section. Based on this, actual sample sizes of 25 in each group provide at least 80% power to detect a 1.0 difference in LCI while actual mean differences were even higher with 1.3 and 1.4 as compared to the healthy and non-healthy control groups, respectively. An estimated 20 percentage point difference was not reached exclusively for D5-20 when comparing non-healthy controls to PH patients. In contrast, there was an actual 35 percentage point difference between healthy controls and PH patients. The “Limitations” section in the discussion was revised accordingly. 

The role of care manager should be discussed. Please consider and discuss the paper from Ciccone MM et al. Vasc Health Risk Manag. 2010 May 6;6:297-305.

A respective paragraph was added to the “Discussion” section as proposed and we want to thank you for introducing this important aspect.

Round  2

Reviewer 1 Report

The authors could not answer comment 1 appropriately.

Author Response

We would like to thank both reviewers for the again expeditious evaluation and helpful comments on our revised manuscript. We adapted the article according to the remaining open point and think this further enhanced its quality. Please find enclosed our point-to-point answers as well as the revised document. Changes introduced during the first revision remain highlighted. Additional adaptions during the second revision are marked yellow to facilitate comparison with previous versions.

Reviewer 1

The authors could not answer comment 1 appropriately. (“Patients with PH included COPD, sarcoidosis, bronchial asthma and fibrosis (Table e3). The data should be analyzed only in group 1 PH.”)  

We respectfully acknowledge the reviewer’s important point and revised the manuscript accordingly. Specifically, a statement on the morphological overlap between lesions found in patients with group 1 and 3 was added to the “Introduction” section. This includes the pulmonary comorbidities such as COPD, bronchial asthma and fibrosis given in Table e3. Being aware of potential confounding, we controlled for this during matching resulting in a balanced distribution of comorbidities in the PH groups and non-healthy controls. Additionally, a statement concerning group 2 PH as the clinically most relevant group was added to the “Discussion” section. These refer to the cardiac comorbidities in Table e3. Taken together, we decided to include all groups of PH as well as comorbidities in our analysis. This gives a realistic representation of patients with PH to be investigated with IOS and MBW in clinical routine as well as in scientific context. However, we are aware that further research is warranted in the distinct subgroups and added a respective statement to the “Limitations” section.

We hope that we were able to address your comment satisfactorily and of course we stand ready to make additional adaptions if required! 

Reviewer 2 Report

To:

Editorial Board

Journal of Clinical Medicine

Title: “Small airway disease in pulmonary hypertension – additional diagnostic value of multiple breath washout and impulse oscillometry”.

Dear Editor,

I read the revised version of this manuscript and I think that the paper is good and well written. The authors had well addressed all my previous comments. The paper improved very much.

Author Response

(The authors gave the same response as above.)
